# Full-Length Transcriptome Profiling of *Coridius chinensis* Mitochondrial Genome Reveals the Transcription of Genes with Ancestral Arrangement in Insects

**DOI:** 10.3390/genes14010225

**Published:** 2023-01-15

**Authors:** Shiwen Xu, Yuange Duan, Ling Ma, Fan Song, Li Tian, Wanzhi Cai, Hu Li

**Affiliations:** Department of Entomology and MOA Key Lab of Pest Monitoring and Green Management, College of Plant Protection, China Agricultural University, Beijing 100193, China

**Keywords:** *Coridius chinensis*, full-length transcriptome, mitochondrial gene transcription, RNA processing

## Abstract

*Coridius chinensis* (Hemiptera: Dinidoridae) is a medicinal insect. Its mitochondrial gene arrangement is consistent with that of *Drosophila melanogaster* and *Erthesina fullo*, the two insects with well-studied mitochondrial transcription. To investigate whether the structural consistency of mitochondrial genes leads to similarities in transcription and post-transcriptional processing, we improved the gene annotation and constructed a quantitative transcription map for the *C. chinensis* mitochondrial genome (mitogenome) using full-length transcriptome sequencing. The size of this mitogenome was 16,214 bp and the proposed model of mitochondrial transcription was similar to that of *Drosophila*. Both strands were nearly entirely transcribed except for the antisense genes downstream of *trnS2* on N strand. The expression of cytochrome c subunit genes is higher than that of NADH-dehydrogenase subunit genes. The post-transcriptional cleavage process followed the “tRNA punctuation” model, and both the “reverse cleavage” model in *Drosophila* and “forward cleavage” model in *E. fullo* were found in *C. chinensis*. In addition, we found that long non-coding RNAs from the control region contained tandem repeats. Polyadenylation was performed after CCA triplet at the 3′ end of tRNA. The isoform diversity of *lrRNA* was identified. Our study sheds light on the transcriptional regulation and RNA processing of insect mitogenomes with the putative ancestral gene arrangement.

## 1. Introduction

The insect mitochondrial genome (mitogenome) is a compact double-strand circular DNA that typically includes 13 protein-coding genes (PCGs), 22 transfer RNA genes (tRNAs), 2 ribosomal RNA genes (rRNAs) [1,2], and a control region (CR) used for regulating the replication and transcription of the mitogenome [1,2,3]. The arrangement of genes in insect mitogenomes is relatively conserved and the arrangement is consistent with the putative ancestral type of *Drosophila melanogaster* [1,4]. Berthier et al. [5] inferred five transcriptional cassettes in *Drosophila* with most antisense genes untranscribed, the finding of which was supported by 5′ and 3′ RACE (rapid amplification of cDNA ends), circularization, and RT-PCR (reverse transcription–polymerase chain reaction) methods [6]. Roberti et al. [7] discovered that *Drosophila* mitochondrial transcription termination factor (DmTTF) bound to *trnE/trnF* and *trnS2/ND1* and proposed two models of mitochondrial transcription. Both strands were almost entirely transcribed in the two models, but in one model, two antisense gene clusters (*astrnS2-asND6* on the J strand and *astrnF-asND4L* on the N strand) were untranscribed.

RNA processing occurs during post-transcriptional processes. A “tRNA punctuation” model for post-transcriptional cleavage of mitochondrial genes was widely accepted [8]. In this model, tRNAs were removed from the primary transcripts, leaving messenger RNA (mRNA) and rRNA transcripts. As a supplement to this model, the “reverse cleavage” model in *Drosophila* [6] and the “forward cleavage” model in *Erthesina fullo* [9] were proposed based on the tRNAs being cleaved from 5′ to 3′ or from 3′ to 5′, respectively. These studies provide a general profile of mitochondrial gene transcription and processing in insects.

Further detailed findings in transcriptional and post-transcriptional regulation were made on the basis of such knowledge. For example, a study of mitochondrial transcription in *E. fullo* [9] described the expression profile of mitochondrial genes and identified 3′ polyadenylation, natural antisense transcripts, and isoform diversity of rRNA. There are many studies on CR due to its important role in transcriptional regulation [10]. Gao et al. [11] reported novel long non-coding RNAs (lncRNAs) transcribed from two regions of CR, defined as Mitochondrial D-loop 1 gene (*MDL1*) and Mitochondrial D-loop 1 antisense gene (*MDL1AS*). This finding contradicts the previous argument that CR is an untranscribed region [12]. On this basis, the *eft-MDL1* gene was defined in CR of *E. fullo*. The *eft-MDL1* gene contains multiple repeat units and repeated transcription initiation sites (TISs), the copy number of which varies within an individual [12]. This copy number variation of short tandem repeats explains the heterogeneity of mitochondrial sequence in individual insect [13].

The above-mentioned transcription and RNA processing patterns of mitochondrial genes were mostly studied in insects such as *Drosophila* and *E. fullo.* Little is known about the RNA processing patterns of mitochondrial genes in other insects with the putative ancestral gene arrangement. *Coridius chinensis* (Hemiptera: Dinidoridae) is a crop pest which is also used as a traditional Chinese medicine to regulate breathing and relieve pain, suggesting its economic and medicinal value [14]. Our previous study showed that the mitogenome of *C. chinensis* included all 37 genes (13 PCGs, 22 tRNAs, and 2 rRNAs) with the ancestral gene arrangement [15]. However, the incomplete CR obtained in our previous study [15] did not allow us to accurately parse the transcriptional regulation. To investigate whether the consistency in gene arrangement leads to similarities in transcription and post-transcriptional processing, we obtained the complete mitogenome and constructed a quantitative transcription map for *C. chinensis* using high-quality full-length transcripts generated from single-molecule, real-time (SMRT) sequencing. The models of mitochondrial transcription and post-transcriptional cleavage in *C. chinensis* were proposed and compared with those of *D. melanogaster* and *E. fullo*. Further detailed findings including lncRNAs from CR with tandem repeats, 3′ end polyadenylation of tRNAs, and isoform diversity of *lrRNA* were also obtained. These results provide insight into the formation and function of unique transcripts, patterns of transcription, and transcriptional regulation of insect mitogenomes with the putative ancestral gene arrangement.

## 2. Materials and Methods

### 2.1. Full-Length Transcriptome Sequencing

Specimens of *C. chinensis* were collected from Zhanggou Village, Leshan City, Sichuan Province, China (29.51° N 103.42° E), in November 2021. Voucher specimens for sequencing were preserved under −80 °C at the Entomological Museum of China Agricultural University, Beijing, China. One male adult and one female adult of *C. chinensis* were pooled for total RNA extraction, using TRIzol Universal (Tiangen, Beijing, China). The concentration and purity of total RNA was detected by NanoDrop 2000 spectrophotometry (Thermo Fisher, Waltham, MA, USA), and the integrity was detected by the Agilent 2100 system (Agilent Technologies, Santa Clara, CA, USA). Total RNA was reverse transcribed into cDNA using the Clontech SMARTer PCR cDNA Synthesis Kit (Clontech Laboratories, Inc., Mountain View, CA, USA) with 3′ SMART CDS Primer II A (5′-AAGCAGTGGTATCAACGCAGAGTAC-T(30)-3′) and SMARTer II A Oligonucleotide (5′-AAGCAGTGGTATCAACGCAGAGTACATGGG-3′). The cDNA was amplified and the library with the insert size between 1 and 10 kb was constructed according to the PacBio Iso-Seq protocol after size selection. Finally, one SMRT cell was performed on the Pacbio Sequel platform with circular consensus sequencing (CCS) mode at Berry Genomics Company (Beijing, China).

### 2.2. Data Assessment and Quality Control of RNA Raw Data

The PacBio SMART Analysis v10.2 (http://www.pacb.com/devnet/, accessed on 28 November 2022) was performed on the raw data. The sequenced reads were processed into the high-quality circular consensus sequencing (CCS) reads using CCS v5.0.0 with parameters (minimum full passes = 1, minimum predicted accuracy = 0.9). Then, Lima v2.0.0 was used to produce full-length non-chimera (FLNC) reads known as draft transcripts by removing the full-length chimera, 5′ and 3′ ends of cDNA primers, and 3′ ployA sequences. The BAM file was converted to a FASTA file using SAMtools v1.11 (http://www.htslib.org/ (accessed on 9 December 2022)) [16].

### 2.3. Mitogenome Sequencing

The genomic DNA was extracted from one adult individual using the DNeasy Blood and Tissue kit (Qiagen, Dusseldorf, Germany) on the basis of the manufacturer’s protocol. The Illumina TruSeq library was prepared with an average insert size of 350 bp and sequenced with the paired-end reads length of 150 bp on the Illumina NovaSeq 6000 platform (Berry Genomic, Beijing, China). A total of 6 Gb raw data was obtained and the short and low-quality reads were removed by using Prinseq version 0.20.4 [17] with the parameter poly-Ns > 15 bp, or >75 bp bases with quality score < 3. The remaining reads were de novo assembled using IDBA-UD [18], with minimum and maximum k values of 41 and 141 bp, respectively. The corresponding mitogenomic contig was identified by *COX1* sequence (GenBank accession number JQ739179) using BLAST.

The gene sequences were preliminarily annotated by MitoZ [19] with the settings “genetic_code 5” and “clade Arthropoda” and further accurately corrected by alignment with homologous genes of other Dinidoridae species using Geneious v10.2.6 (http://www.geneious.com/ (accessed on 9 December 2022)) [20] with high-sensitivity map and default parameters.

### 2.4. Mitochondrial Transcriptome Analysis

The draft transcripts in FASTA format were mapped to complete the reference mitogenome in linear topology using Geneious. The parameters used for mapping were maximum gap per read = 10%, maximum gap size = 20, minimum overlap identity = 85%, maximum mismatches per read = 10%, and maximum ambiguity = 16. Geneious was used to calculate the coverage of each base in the reference mitogenome to obtain the coverage histogram, namely the quantitative transcription map. The 5′ and 3′ ends of mature transcripts, polycistronic transcripts, and antisense transcripts were identified and classified after precise modification of the mitogenome according to the full-length transcripts. We selected a representative transcript from each of the same type and plotted it on the quantitative transcription map.

## 3. Results

### 3.1. The Annotation of C. chinensis Mitogenome

We annotated the mitogenome of *C. chinensis* using DNA sequence features and modified it precisely according to the full-length transcripts (Table 1). Compared to the annotations using DNA sequence features, five PCGs (*ND2*, *ATP6*, *COX3*, *ND3*, and *ND1*) and one rRNA (*srRNA*) were re-annotated according to the full-length transcripts. The modifications were mainly made on the start and stop codons of PCGs. DNA sequence features suggested that *lrRNA* was located between *trnL2* and *trnV*, and *srRNA* was located between *trnV* and CR, while transcripts of *lrRNA* and *srRNA* have more accurately located the two genes. The transcriptional annotations showed that the 5′ ends of three genes (*ND3*, *ND1*, and *srRNA*) were altered, resulting in the change in start codon in PCGs. The modified 5′ ends were still initiated at the start codon of PCGs and did not affect the translation. The 5′ end of *srRNA* was confirmed by 273 draft transcripts for fidelity, and the location of CR was determined accordingly. The 3′ ends of three genes (*ND2*, *ATP6*, and *COX3*) were altered, leaving T or TA residue as incomplete stop codons. The residue was completed by polyadenylation, as in other studies [6,12]. The overlapping regions between genes *ND2* and *trnW*, *ATP6* and *COX3*, *COX3* and *trnG*, *trnG* and *ND3*, and *ND1* and *trnL1* disappeared, and therefore, the mitogenome became more compact.

Based on the precise annotations according to the full-length transcripts, the complete mitogenome (GenBank accession number OP921229) of *C. chinensis* was 16,214 bp in size and included all 37 genes (13 PCGs, 22 tRNAs, and 2 rRNAs). The arrangement of these genes in *C. chinensis* was consistent with the putative ancestral arrangement. The complete CR was 1553 bp in length and contained tandem repeated units. The repeat region contained a 19 bp sequence (type I) tandemly repeated 15 times, an 18 bp sequence (type II) repeated 10 times, and a 37 bp sequence (type III) repeated 7 times with a partial 35 bp repeat.

### 3.2. The Quantitative Transcription Map of C. chinensis Mitogenome

We set out to quantify the gene expression profile of the *C. chinensis* mitogenome. We obtained a total of 20.456 Gb raw data with 18,776,321 subreads by full-length transcriptome sequencing. The subread N50 is 1226 bp, N90 is 2527 bp, and the longest subread length is 257,190 bp. The raw reads generated 247,996 high-quality CCS (accuracy ≥ 0.9) transcripts, and produced 230,030 draft transcripts. In these draft transcripts, 61,118 transcripts were mapped to the complete mitogenome of *C. chinensis* with a mean coverage of 11,289.6 bases. The quantitative transcription map showed that the sense transcripts were much more abundant than antisense transcripts (Figure 1). We counted the numbers of sense transcripts and antisense transcripts separately. We found that the relative expression levels of mitochondrial genes, ranking from highest to lowest, were *lrRNA*, *COX2*, *srRNA*, *COX3*, *CYTB*, *ND1*, *COX1*, *ATP6*, *ATP8*, *ND5*, *ND2*, *ND6*, *ND4*, *ND3*, and *ND4L*. The antisense genes *asND6*, *asCYTB*, *asND5*, *asND2*, and *asND4* were highly expressed.

### 3.3. Detection of lncRNAs from Mitochondrial Control Region

LncRNAs from CR were considered to be the precursors of the transcription initial RNAs (tiRNAs) [11], which were close to the promoters [21] and can indicate TISs. We found a total of 14 transcripts concentrated in the CR of the *C. chinensis* mitogenome, including polycistronic transcripts (Appendix A). Four lncRNAs were transcribed from N strand and two of them (see nos. 5243673 and 95946640) initiated at position 15,377. In addition, the incomplete polycistrons *CR/trnI* (see no. 119408095; ChrM: 15,378-45) and *CR/trnI/astrnQ/trnM/ND2* (see no. 127142199; ChrM: 15,379-408) indicated the presence of primary transcripts covering CR to *ND2*. Therefore, the TIS of the N strand may be located at position 15,377 in CR. Ten lncRNAs were transcribed from the J strand. The 3′ end of transcript *astrnI/CR* (see no. 90702191; ChrM: 72-15,406) was located in CR and was considered to be in close vicinity of the transcription termination site (TTS) at position 15,406. Of the remaining nine transcripts, five were only concentrated in CR and four covered *srRNA* and CR. The 5′ ends of these transcripts were downstream of position 15,189, with the exception of two transcripts (see nos. 97583892 and 138412294; ChrM: 15,189-14,901 and 15,189-12,846) initiated at position 15,189, which may be the TIS of the J strand.

In addition, we detected five lncRNAs from CR which contained tandem repeat sequences. Four of these lncRNAs were transcribed from the N strand and one was transcribed from the J strand (Figure 2). The repeat region in lncRNAs was downstream of TIS of the N strand and upstream of TIS of the J strand. The TISs of both strands were not in the repeat region. The repeat region initiated at position 15,464 and was composed of three different types of repeat units. The repeat region of transcript no. 5243673 contained a 19 bp sequence (type I) tandemly repeated 17 times, an 18 bp sequence (type II) tandemly repeated 3 times, and a 37 bp sequence (type III) tandemly repeated 2 times with a partial third 33 bp repeat. In the other four sequences, only type I repeat units were tandemly repeated, while type II and type III repeat units were interspersed. There were four polymorphic sites with alleles CTA/TTA, G/TTA, T/C, and C/T in type I of repeat units and type III of repeat units, respectively.

### 3.4. The Proposed Model of Mitochondrial Transcription

Two gene clusters *ATP8/ATP6* and *ND4/ND4L* were transcribed into polycistronic transcripts and formed mature mRNAs, while the other mature RNAs were monocistronic transcripts. Furthermore, some polycistronic transcripts, antisense transcripts, and lncRNAs from CR were found and considered as the primary transcripts or the products of post-transcriptional processing. There were 175 polycistronic transcripts (except for *ATP8/ATP6* and *ND4/ND4L*), antisense transcripts, and lncRNAs from CR mapped to the mitogenome of *C. chinensis*, including 72 transcripts transcribed from N strand and 103 transcribed from J strand (Appendix A). All of the sense gene clusters and three antisense gene clusters (*astrnE*-*asCOX1* on J strand, *astrnS2-asND6* on J strand, and *astrnF-asND4L* on N strand) were transcribed. However, the absence of transcripts downstream of *trnS2* suggested that the antisense genes downstream of *trnS2* of N strand were not transcribed and that there may be a TTS at *trnS2/ND1*.

Based on the above analyses, we proposed a transcriptional model of the *C. chinensis* mitogenome (Figure 3). Both J and N strands contain at least one TIS and one TTS, respectively. Except for the TTS of N strand at *trnS2/ND1*, the TISs of both strands and the TTS of the J strand were all found in CR. The mitogenome was transcribed into an almost whole-genome primary polycistron, and only the antisense genes downstream of *trnS2* of the N strand was not transcribed. However, further research was needed to determine the presence and exact location of TISs and TTSs.

### 3.5. Transcripts of tRNA and rRNA

We obtained 26 tRNAs transcribed from nine tRNA genes in *C. chinensis*. Among these, 21 were transcribed from the N strand and 5 from the J strand (Appendix A), and no antisense transcript of tRNA genes was found. Among the 26 tRNAs, 15 had complete transcripts and 11 were incomplete transcript genes. Only one of the incomplete tRNAs was degraded at the 3′ end and the other 10 were degraded at the 5′ end.

We found that polycistronic transcripts *trnM/ND2*, *trnL2/COX2*, *trnG/ND3/trnA/trnR*, and *trnG/ND3* lack tRNAs downstream of mRNAs, indicating that they had the ability to remove tRNAs by 3′ to 5′ cleavage. However, transcript of *trnL1* downstream of *lrRNA* was observed in the polycistronic transcript *lrRNA/trnL1*, indicating that the *trnV* upstream of *lrRNA* was removed by 5′ to 3′ cleavage. Except for the incomplete transcript (no. 108265960) which was degraded at the 3′ end with the loss of CCA, none of the CCA triplets found at the 3′ end of tRNAs were encoded by the mitogenome of *C. chinensis* but were post-transcriptionally synthesized by tRNA nucleotidyltransferase, as reported in previous studies [21]. In addition to canonical cleavage processing and post-transcriptional modification of primary transcripts of tRNAs, we also found polyadenylation of tRNAs after the addition of CCA triplet.

There were 58,809 monocistronic RNA reads aligned to the *lrRNA* gene and 452 aligned to the *srRNA* gene. The quantitative transcription map showed that the number of transcripts degraded at the 3′ end was much lower than that of the 5′ end of *lrRNA*, while the opposite was seen for *srRNA* (Figure 1). In addition, three isoforms of *lrRNA* (ChrM: 13,687-12,501, 13,498-12,501, and 13,484-12,501) were confirmed by at least 4000 draft transcripts, and the 3′ end of these transcripts was located at the 3′ end of the *lrRNA* gene, while the 5′ end was located in the *lrRNA* gene.

## 4. Discussion

Compared with our previous study on mitogenome of *C. chinensis* [15], this current work obtained the complete CR sequence. We studied the tandem repeated units in CR and modified the annotations of genes precisely. This laid the foundation for the subsequent study of the mitogenome of *C. chinensis* at the transcriptional level. The ubiquitous presence of polyA tails at the 3′ end of transcripts often prevents us from accurately annotating whether the exact 3′ ends were transcribed from the stop codon or added after transcription [6,22,23]. For example, the A residues of the stop codon of *ND2* and *COX3* may either be transcribed or added post-transcriptionally following cleavage after the first position U.

By sequencing the full-length transcriptome, we constructed a quantitative mitochondrial transcription map of *C. chinensis*. The cytochrome c (*COX*) subunits constantly had higher expression levels than NADH-dehydrogenase (*ND*) subunits in human and insects [5,9,11], suggesting that the transcription levels of these genes were highly conserved across metazoans. However, the relative expression level of each gene was not exactly consistent in insects [5,9]. Compared with the levels obtained by densitometric analysis in *Drosophila* [5], the gene expression levels in *C. chinensis* showed a certain gradient. The expression levels of *srRNA*, *ND1*, *COX1*, *ND5*, and *ND4/ND4L* were different between *C. chinensis* and *E. fullo* [9]. We then studied the corresponding antisense genes in the two species and found that *assrRNA*, *asND1, asCOX1*, and *asND4/asND4L* were expressed at consistent levels. This suggested that interspecific differences in gene expression levels were not significantly correlated with the expression of antisense transcripts in our study. Previous studies in *Drosophila* showed that the polyA tails of *srRNA* transcripts were relatively short, which were not efficiently captured by the OligoT binding methodology [6], explaining the much lower expression level of *srRNA* than *lrRNA* in the quantitative transcription map. In *C. chinensis* and *E. fullo*, however, the lengths of polyA tails and the polyadenylation status were similar between *lrRNA* and *srRNA* [9]. Thus, in these two species, the polyadenylation status did not seem to explain the higher expression of *lrRNA* than *srRNA*. However, our study on gene expression was based on long-read sequencing and lacked molecular experiments such as qPCR (quantitative real-time polymerase chain reaction). Further experiments are needed to confirm whether these mitochondrial genes indeed exhibit the expression profiles as observed by the long-read sequencing.

The models of transcription and the post-transcriptional cleavage process were proposed in the mitogenome of *C. chinensis*. The proposed transcriptional mode of *C. chinensis* was similar to that of the model organism *Drosophila* [7]. Both strands were almost entirely transcribed except for the non-coding region (NCR) downstream *trnS2* on the N strand. The significant reduction in the number of transcripts at *trnS2/ND1* of both strands suggests that there may be a mitochondrial transcription termination factor (mTTF) binding site at *trnS2/ND1* in *C. chinensis*. The mTTF would only serve as an attenuator of the J strand but as a true terminator of the N strand because no transcripts were observed downstream of the protein-binding site. The location of this binding site is consistent with DmTTF in *Drosophila*, whereas another DmTTF binding site at *trnE/trnF* was not reflected in the transcription map of *C. chinensis*. The post-transcriptional cleavage process followed the “tRNA punctuation” model [8] in *C. chinensis*, which was similar to the patterns observed in other insects [6,9,24]. The cleavage process of four transcripts, *trnM/ND2, trnL2/COX2*, *trnG/ND3/trnA/trnR*, and *trnG/ND3*, followed the “reverse cleavage” model proposed in *Drosophila* [6]. The cleavage process of the transcript *lrRNA/trnL1* followed the “forward cleavage” model, which was raised in *E. fullo* [9]. In addition, the cleavage patterns of mRNAs were slightly different across insects. The mature bicistronic transcripts *ND6/CYTB* were found in the mitochondrial transcriptome of *Bemisia tabaci* and *Maruca vitrata*, and the tricistronic transcripts *ATP8/ATP6/COX3* were found in *M. vitrata* [24,25]. However, in *Drosophila, E. fullo*, and *C. chinensis*, the bicistronic transcripts *ND6/CYTB* were cleaved into monocistronic transcripts, and the tricistronic transcripts *ATP8/ATP6/COX3* were present as separate *ATP8/ATP6* and *COX3* transcripts despite the lack of an intervening tRNA.

After being cleaved from the primary transcripts, the CCA triplet was added to the 3′ end of the tRNA by template-independent editing reaction using tRNA nucleotidyltransferase 1 (TRNT1) [22]. As in most eukaryotes, endonucleases were used for tRNA 3′ end maturation in mitochondria of *C. chinensis* [26]. No precursor of tRNAs supporting the use of exonuclease was found in *C. chinensis*. Processing of tRNA 3′ ends, including CCA addition, was considered to be an efficient process to compete with polyadenylation [27]. However, polyadenylation following the addition of CCA triplet at the 3′ end of tRNA was observed in our study and others [22,28]. This phenomenon was described as aberrant processing, the biological significance of which is yet to be determined [28]. It was reported in *Saccharomyces cerevisiae* that pre-tRNA_i_Met was degraded by Rrp6 and the nuclear exosome, after polyadenylation by Trf4 [29]. Many incomplete tRNAs were found to have incomplete 5′ ends but perfect 3′ ends, and followed by CCA triplets and polyA tails in turn. These observations suggest that polyadenylation might be related to the degradation of mature tRNAs. In *E. fullo*, the *srRNA* gene was transcribed in three isoforms [9], whereas isoforms only existed in the transcript of *lrRNA* in our study. There are two exonucleolytic degradation mechanisms of RNA, 5′ to 3′ and 3′ to 5′ exonucleolytic degradation [30]. Therefore, we hold the opinion that the isoforms were due to exonucleolytic degradation, and the end of isoforms may be an indication of splice site in exonucleolytic degradation. In addition, the trapezoid formed by the long to short transcripts in the quantitative transcription map indicated that 5′ to 3′ exonucleolytic degradation was dominant in *lrRNA* and 3′ to 5′ exonucleolytic degradation was dominant in *srRNA* in *C. chinensis*. Although sequencing errors could not be completely ruled out for these incomplete transcripts, since they were confirmed by multiple reads, we believe that they truly reflect the sequence information. The incomplete RNAs in degradation may not be able to perform normal physiological functions.

Tandem repeats in CR were found in the mitogenome in an increasing number of species [31,32]. In previous studies, copy number variation of tandem repeats has been found to affect the length of the CR [33]. Due to the potentially deleterious effect of tandem repeats, their occurrence should be constrained by natural selection [34]. However, a study based on full-length transcriptome proposed that the TISs were in the repeat unit and repeated more than 10 times in *E. fullo* [12]. The hypothesis was that TISs could prevent deleterious mutation through tandem repeats. In our study, we found five tandem repeat patterns in lncRNAs from CR, but there was no inferred TIS or TTS in these repeat units. The copy number and pattern of tandem repeats showed strong heterogeneity. We speculate that this may follow the tandem duplication random loss (TDRL) model [35], with duplication of a portion of the DNA followed by the potential random loss of supernumerary DNA in tandem duplication, resulting in different tandem repeat patterns.

## 5. Conclusions

In summary, insect mitogenomes with the putative ancestral gene arrangement have many common characteristics in the expression levels of mitochondrial genes, the models of transcription and post-transcriptional cleavage processes, and the formation and function of unique transcripts. Our findings shed light on mitochondrial gene transcription, RNA processing, and RNA degradation in insect mitogenomes with the putative ancestral gene arrangement.

## Figures and Tables

**Figure 1 genes-14-00225-f001:**
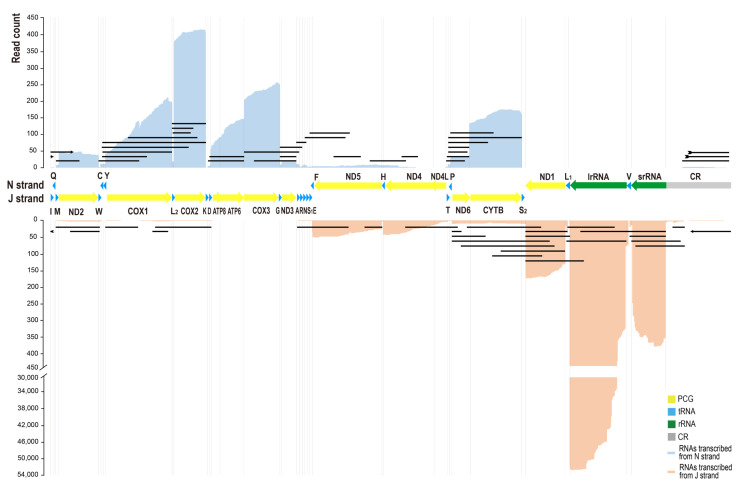
The quantitative transcription map of *C. chinensis* mitogenome. The mitogenome was arranged in the J strand orientation. Alignments of transcripts encoded by J strand were piled along the positive y-axis in blue color. Alignments of transcripts encoded by N strand were piled along the negative y-axis in orange color. The black lines represent types of polycistronic transcripts, antisense transcripts, and long non-coding RNAs (lncRNAs) from control regions transcribed from N strand (**above**) and J strand (**below**). The lines with arrows indicate the same transcript across the control region.

**Figure 2 genes-14-00225-f002:**
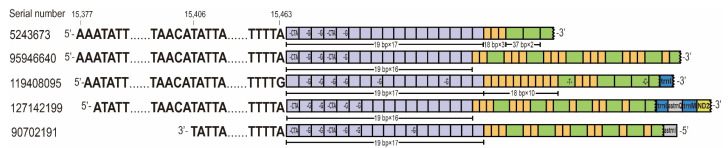
Five lncRNAs from CR contained tandem repeats in *C. chinensis*. Four transcripts (**above**) were encoded by J strand and one (**below**) by N strand. The purple, orange, and green squares represent each of the three tandem repeats and the bases within the squares represent polymorphic sites.

**Figure 3 genes-14-00225-f003:**
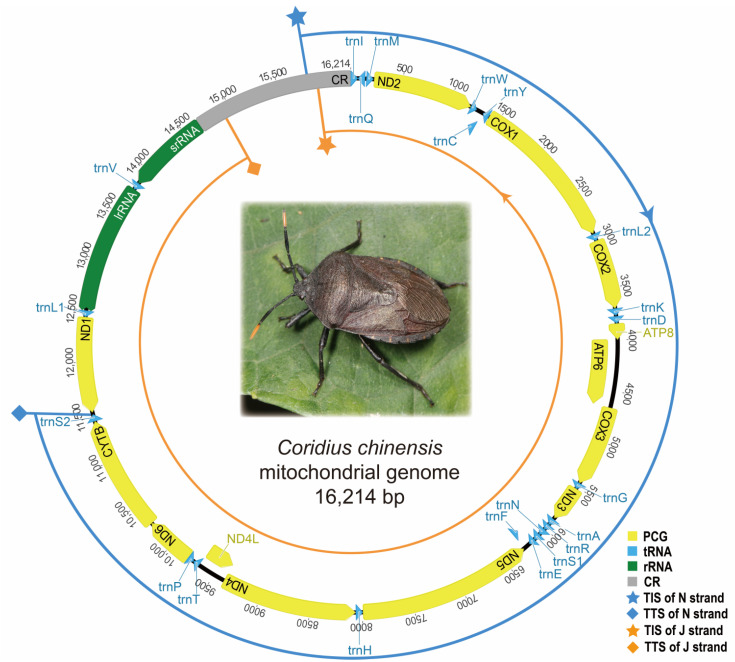
The proposed models of mitochondrial transcription in *C. chinensis*. The blue and orange curves represent the polycistronic primary transcripts of N and J strands, respectively. The orientation of transcription is shown by the arrows.

**Table 1 genes-14-00225-t001:** The precise annotation of the *C. chinensis* mitogenome based on DNA sequencing and full-length transcripts. Abbreviations: *ATP6* and *ATP8*, adenosine triphosphate (ATP) synthase subunits 6 and 8; *COX1*–*COX3*, cytochrome oxidase subunits 1–3; *CYTB*, cytochrome b; *ND1–6* and *ND4L*, NADH-dehydrogenase subunits 1–6 and 4L; *srRNA* and *lrRNA*, large and small rRNA subunits. Transfer RNAs are represented by their amino acid abbreviation of the corresponding amino acid, for transfer RNA (L1: CUN; L2: UUR; S1: AGN; S2: UCN); CR, control region.

Gene	Coding Strand	Annotations Using DNA Sequencing				Annotations According to Transcripts			
		Position (J Strand as Reference)	Length/bp	Start Codon	Stop Codon	Position (J Strand as Reference)	Length/bp	Start Codon	Stop Codon
*trnI*	J	1-71	71			1-71	71		
*trnQ*	N	142-73	70			142-73	70		
*trnM*	J	155-224	70			155-224	70		
*ND2*	J	229-1206	978	AUA	UAA	229-1,204	976	AUA	U
*trnW*	J	1205-1268	64			1205-1268	64		
*trnC*	N	1329-1261	69			1329-1261	69		
*trnY*	N	1402-1337	66			1402-1337	66		
*COX1*	J	1407-2942	1536	UUG	UAA	1407-2942	1536	UUG	UAA
*trnL_2_*	J	2949-3019	71			2949-3019	71		
*COX2*	J	3020-3698	679	AUA	U	3020-3698	679	AUA	U
*trnK*	J	3699-3770	72			3699-3770	72		
*trnD*	J	3775-3842	68			3775-3842	68		
*ATP8*	J	3844-4002	159	UUG	UAA	3844-4002	159	UUG	UAA
*ATP6*	J	3996-4679	684	AUG	UAA	3996-4665	670	AUG	U
*COX3*	J	4666-5454	789	AUG	UAA	4666-5453	788	AUG	UA
*trnG*	J	5454-5518	65			5454-5518	65		
*ND3*	J	5516-5872	357	AUA	UAA	5519-5872	354	AUA	UAA
*trnA*	J	5881-5949	69			5881-5949	69		
*trnR*	J	5950-6017	68			5950-6017	68		
*trnN*	J	6019-6084	66			6019-6084	66		
*trnS_1_*	J	6085-6153	69			6085-6153	69		
*trnE*	J	6154-6218	65			6154-6218	65		
*trnF*	N	6282-6217	66			6282-6217	66		
*ND5*	N	7990-6290	1701	AUU	UAA	7990-6290	1701	AUU	UAA
*trnH*	N	8057-7992	66			8057-7992	66		
*ND4*	N	9384-8059	1326	AUG	UAG	9384-8059	1326	AUG	UAG
*ND4L*	N	9665-9378	288	UUG	UAA	9665-9378	288	UUG	UAA
*trnT*	J	9668-9732	65			9668-9732	65		
*trnP*	N	9797-9734	64			9797-9734	64		
*ND6*	J	9801-10,280	480	AUG	UAA	9801-10,280	480	AUG	UAA
*CYTB*	J	10,282-11,418	1137	AUG	UAA	10,282-11,418	1137	AUG	UAA
*trnS_2_*	J	11,421-11,489	69			11,421-11,489	69		
*ND1*	N	12,439-11,510	930	AUG	UAA	12,433-11,510	924	AUG	UAA
*trnL_1_*	N	12,500-12,434	67			12,500-12,434	67		
*lrRNA*	N	13,774-12,501	1274			13,774-12,501	1274		
*trnV*	N	13,841-13,775	67			13,841-13,775	67		
*srRNA*	N	14,652-13,842	811			14,661-13,842	820		
CR		14,653-16,214	1562			14,662-16,214	1553		

## Data Availability

The data that support the findings of this study will be available in GenBank at https://www.ncbi.nlm.nih.gov/ (accessed on 9 December 2022). The complete mitogenome of *C. chinensis* was deposited in the NCBI Nucleotide database with accession number OP921229. The raw data of the full-length transcriptome were deposited in the NCBI SRA database with accession number SRR22456283.

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
