# Peer review of "Full-Length Transcriptome Profiling of Coridius chinensis Mitochondrial Genome Reveals the Transcription of Genes with Ancestral Arrangement in Insects"

_genes, 2023, doi:10.3390/genes14010225_

Round 1
Reviewer 1 Report
This manuscript provide a good finding in molecular biology on insect field. The data analysis sound scientifically and the method section has been well explained. The quality of presentation can be acceptable. However, the introduction and discussion sections can be improve due to not well explain and repetition statements occurred. It should be clear and comprehensive. I am suggest minor revision with revise these 2 section to be more comprehensible and understandable.
Author Response
Dear Ms. Bianca Laura Chesca,
We are resubmitting the revised manuscript (genes-2121303) entitled “Full-length transcriptome profiling of Coridius chinensis mitochondrial genome reveals the transcription of genes with ancestral arrangement in insects” for your consideration for publication in Genes.
On behalf of all authors, I would like to thank you and the three reviewers for your valuable comments that have helped us improve this manuscript greatly. We have revised the manuscript thoroughly according to these comments. I detail our responses below.
Reviewer #1 Comments
- This manuscript provide a good finding in molecular biology on insect field. The data analysis sound scientifically and the method section has been well explained. The quality of presentation can be acceptable. However, the introduction and discussion sections can be improve due to not well explain and repetition statements It should be clear and comprehensive. I am suggest minor revision with revise these 2 section to be more comprehensible and understandable.
RESPONSE: We have re-written the introduction and discussion sections according to the reviewer’s comments.
Again, we thank you and the three reviewers for the invaluable comments that have greatly improved our manuscript!
Yours sincerely,
Hu Li
Professor, Ph.D.
Reviewer 2 Report
The study has been well designed and detailed analysis has been performed on full length transcript data from mitochondrial genome of Coridius chinensis. However, I would like to suggest minor improvements to this study.
1. The expression levels of mitochondrial genes in this study do not match with the observed pattern in other insects. Can this be explained by the expression of antisense transcripts in your study. Do the transcripts with corresponding antisense transcripts seem to be showing lower expression in this system than that observed in other insects? Could you check for presence/absence of specific antisense transcripts in other systems? The authors can also discuss any other relevant observations like polyadenylation status of these genes etc to discuss their findings. Alternatively, qPCR can also be performed to confirm whether these mitochondrial genes indeed demonstrate the expression profile as observed by the long read sequencing.
2. The polyadenylation status of the mitochondrial genes observed in this system can be discussed in detail. How does it compare with that found in other insect systems. Especially because there is a lot of ambiguity regarding the same in different systems.
3. I could find a study on mitochondrial gene expression profiling in Maruca vitrata (Lepidoptera) (Margam et al., 2011). Please compare your findings in this or other relevant insect systems to draw parallels or contrasts for better understanding of insect mitochondrial systems.
4. I could also find your previous study on mitochondrial genome of Coridius chinensis. Notable improvements/deviations in the current assembly can be provided in this manuscript.
I am also listing some minor corrections in the manuscript:
Line 16: The placement of the line ' The expression of cytochrome c.. ' looks erratic. It can be brought after Line 19.
Line 35: needs rephrasing
Line 53: use of 'these knowledges' is incorrect.
Line 304: needs rephrasing
Line 323-324: is incomplete
Author Response
Dear Ms. Bianca Laura Chesca,
We are resubmitting the revised manuscript (genes-2121303) entitled “Full-length transcriptome profiling of Coridius chinensis mitochondrial genome reveals the transcription of genes with ancestral arrangement in insects” for your consideration for publication in Genes.
On behalf of all authors, I would like to thank you and the three reviewers for your valuable comments that have helped us improve this manuscript greatly. We have revised the manuscript thoroughly according to these comments. I detail our responses below.
Reviewer #2 Comments
- The expression levels of mitochondrial genes in this study do not match with the observed pattern in other insects. Can this be explained by the expression of antisense transcripts in your study. Do the transcripts with corresponding antisense transcripts seem to be showing lower expression in this system than that observed in other insects? Could you check for presence/absence of specific antisense transcripts in other systems? The authors can also discuss any other relevant observations like polyadenylation status of these genes etc to discuss their findings. Alternatively, qPCR can also be performed to confirm whether these mitochondrial genes indeed demonstrate the expression profile as observed by the long read sequencing.
RESPONSE: We have checked the presence/absence of specific antisense transcripts in other systems and have revised the text about the expression level of antisense transcripts in the results and discussion sections. The polyadenylation status of representative genes of C. chinensis and other insects has also been discussed. We thank reviewer for the suggestion about the qPCR validation which is helpful for our future study in this area, and we have discussed this limitation.
- The polyadenylation status of the mitochondrial genes observed in this system can be discussed in detail. How does it compare with that found in other insect systems. Especially because there is a lot of ambiguity regarding the same in different systems.
RESPONSE: Most previous studies focus on the polyadenylation status in two representative genes (lrRNA and srRNA) to explain the differences in their expression levels. We have compared and discussed the polyadenylation status of these two genes with other insect systems.
- I could find a study on mitochondrial gene expression profiling in Maruca vitrata (Lepidoptera) (Margam et al., 2011). Please compare your findings in this or other relevant insect systems to draw parallels or contrasts for better understanding of insect mitochondrial systems.
RESPONSE: We have compared our results with other insect systems (e.g., Maruca vitrata, Bemisia tabaci, Drosophila and Erthesina fullo) in discussion section.
- I could also find your previous study on mitochondrial genome of Coridius chinensis. Notable improvements/deviations in the current assembly can be provided in this manuscript.
RESPONSE: We added the relevant content to the discussion section.
- Line 16: The placement of the line ' The expression of cytochrome c.. ' looks erratic. It can be brought after Line 19.
RESPONSE: Have corrected.
- Line 35: needs rephrasing
RESPONSE: Have rephrased.
- Line 53: use of 'these knowledges' is incorrect.
RESPONSE: We have changed the phrase into ‘such knowledge’.
- Line 304: needs rephrasing
RESPONSE: Have rephrased.
- Line 323-324: is incomplete
RESPONSE: Have corrected.
Again, we thank you and the three reviewers for the invaluable comments that have greatly improved our manuscript!
Yours sincerely,
Hu Li
Professor, Ph.D.
Reviewer 3 Report
The manuscript by Xu et al. titled Full-length transcriptome profiling of Coridius chinensis mitochondrial genome reveals the transcription of genes with ancestral arrangement in insects is an interesting and well-written paper. The authors however need to make a few minor corrections to enhance the work.
Abstract
The length/size of the mitogenome should be stated in the abstract.
Apart from universally-accepted abbreviations such as DNA, RNA, etc, others such as RACE (Rapid amplification of cDNA ends) should be defined in full at first mention.
The authors should consistently italicise all genes (including in the tables) in the manuscript.
Introduction
Citation [15] is not appropriate for the utility of the insect, thus its economic and medicinal importance. The citation [15] is on work done on the mitogenome of the insect so the authors should find an appropriate citation/reference for the economic and medicinal importance of the insect. The authors should also draw a linkage between their previous work [15] and the justification for this current work.
Materials and Methods
Was a voucher specimen kept? Since some of the authors have published a paper on the mitogenome of C. chinensis collected from a different location (citation [15]) and which has a different mitogenome size, it will be good if there was a voucher specimen of the collection used in this current study.
Were the male and female pooled for the total RNA extraction?
All kits/devices/equipment used in the study should have the name of the manufacturer and the place/country of manufacture provided.
Settings of the parameters for the alignment should be stated, if they vary from the default settings (lines 120-122).
The quality of the image in Fig. 3 should be improved.
Discussion
Generally, the authors did a very good job. However, they should also relate the findings of their previous work [15] with that of the current study, particularly on the mitogenome component. They should do this by indicating the similarities and pointing out the differences, if any, and their significance/relevance.

Author Response
Dear Ms. Bianca Laura Chesca,
We are resubmitting the revised manuscript (genes-2121303) entitled “Full-length transcriptome profiling of Coridius chinensis mitochondrial genome reveals the transcription of genes with ancestral arrangement in insects” for your consideration for publication in Genes.
On behalf of all authors, I would like to thank you and the three reviewers for your valuable comments that have helped us improve this manuscript greatly. We have revised the manuscript thoroughly according to these comments. I detail our responses below.
Reviewer #3 Comments
- The length/size of the mitogenome should be stated in the abstract.
RESPONSE: We have added the length/size of the mitogenome in the abstract.
- Apart from universally-accepted abbreviations such as DNA, RNA, etc, others such as RACE (Rapid amplification of cDNA ends) should be defined in full at first mention.
RESPONSE: We have mentioned the full names of the abbreviations at their first appearance.
- The authors should consistently italicise all genes (including in the tables) in the manuscript.
RESPONSE: Have corrected accordingly.
- Citation [15] is not appropriate for the utility of the insect, thus its economic and medicinal importance. The citation [15] is on work done on the mitogenome of the insect so the authors should find an appropriate citation/reference for the economic and medicinal importance of the insect. The authors should also draw a linkage between their previous work [15] and the justification for this current work.
RESPONSE: We have added the new reference for the economic and medicinal importance of this species, also have drawn a linkage between previous and current work.
- Was a voucher specimen kept? Since some of the authors have published a paper on the mitogenome of C. chinensis collected from a different location (citation [15]) and which has a different mitogenome size, it will be good if there was a voucher specimen of the collection used in this current study.
RESPONSE: We have preserved the voucher specimen at the Entomological Museum of China Agricultural University, Beijing, China.
- Were the male and female pooled for the total RNA extraction?
RESPONSE: Yes, we pooled one male and one female adult for total RNA extraction.
- All kits/devices/equipment used in the study should have the name of the manufacturer and the place/country of manufacture provided.
RESPONSE: Have provided accordingly.
- Settings of the parameters for the alignment should be stated, if they vary from the default settings (lines 120-122).
RESPONSE: We have added the settings and parameters.
- The quality of the image in Fig. 3 should be improved.
RESPONSE: Have improved.
- Generally, the authors did a very good job. However, they should also relate the findings of their previous work [15] with that of the current study, particularly on the mitogenome component. They should do this by indicating the similarities and pointing out the differences, if any, and their significance/relevance.
RESPONSE: We have compared the mitogenome component to the precious work, especially the improvement of the gene annotation and the structural features of the complete control region.
Again, we thank you and the three reviewers for the invaluable comments that have greatly improved our manuscript!
Yours sincerely,
Hu Li
Professor, Ph.D.